# A Role of Coagulant in Structure Formation of Fibers and Films Spun from Cellulose Solutions

**DOI:** 10.3390/ma13163495

**Published:** 2020-08-07

**Authors:** Valery Kulichikhin, Igor Makarov, Maria Mironova, Lyudmila Golova, Markel Vinogradov, Georgiy Shandryuk, Ivan Levin, Natalia Arkharova

**Affiliations:** 1A.V. Topchiev Institute of Petrochemical Synthesis, Russian Academy of Sciences, 29, Leninsky prospekt, 119991 Moscow, Russia; makarov@ips.ac.ru (I.M.); mariamir@inbox.ru (M.M.); glk@ips.ac.ru (L.G.); vin1989@ips.ac.ru (M.V.); gosha@ips.ac.ru (G.S.); levin@ips.ac.ru (I.L.); 2A.V. Shubnikov Institute of Crystallography, Federal Research Center Crystallography and Photonics, Russian Academy of Sciences, 119333 Moscow, Russia; natalya.arkharova@yandex.ru

**Keywords:** cellulose, *N*-methylmorpholine-*N*-oxide, fibers spinning, films formation, coagulation, optical interferometry, phase diagram, structure, morphology

## Abstract

Replacing the aqueous coagulation bath with an alcoholic one during spinning cellulose fibers (films) from solutions in *N*-methylmorpholine-*N*-oxide leads to a radical restructuring of the hydrogen bonds net of cellulose and, as a result, to a change in the structure and properties of the resulting material. By the method of optical interferometry, it was possible to identify the intrinsic features of the interaction of the solvent and isomeric alcohols and to construct phase diagrams of binary systems describing the crystalline equilibrium. Knowledge of the phase states of the system at different temperatures renders it possible to exclude the process of solvent crystallization and conduct the spinning in pseudo-homogeneous conditions. The structure and morphology of samples were studied using X-ray diffraction and scanning electron microscopy methods for a specific coagulant. When the solution under certain conditions is coagulated at contact with alcohol, the solvent may be in a glassy state, whereas, when at coagulation in water, an amorphous-crystalline structure is formed. The structural features of cellulose films obtained by coagulation of solutions with water and alcohols help to select potential engineering or functional materials (textile, packaging, membranes, etc.), in which their qualities will manifest to the best extent.

## 1. Introduction

Since cellulose is actually a non-melting polymer, it is necessary to dissolve it in order to obtain molded products (spun fibers, films, etc.), i.e., to transfer it to a liquid state. The main problem limiting the solubility of cellulose is the presence in the glucopyranose unit of hydroxyl groups, initiating a dense system of intra- and intermolecular H-bonds (Figure 1), which prevents the transfer of individual macromolecules to the solution under the action of even strong aprotic solvents.

In native cellulose (polymorphic form I), the distances between the intramolecular hydrogen bonds O(3) H⋯O(5) and O(2) H⋯O(6) are 0.275 and 0.287 nm, respectively; the intermolecular H-bonds O(6) H⋯O(3) have a length of 0.279 nm [1]. The average length of hydrogen bonds for regenerated cellulose (polymorphic form II or hydrate-cellulose) is 0.272 nm [2]. This difference in the lengths of hydrogen bonds makes hydrate-cellulose thermodynamically more stable compared to native cellulose, and it is cellulose materials with this structure that are used in the most popular materials (viscose fibers and cellophane).

Thus, hydrogen bonds are responsible for the restrictions imposed on the methods of processing cellulose, namely excluding thermal methods and limiting the number of its potential direct solvents [3,4]. It follows that the dissolution of cellulose requires solvents with high donor activity, which can destroy the system of H-bonds of cellulose. Note that we are talking about direct cellulose solvents and not about substances that dissolve cellulose esters and ethers, as is the case in the popular viscose process [5].

At the beginning of the 1970s, it appeared and began to be actively introduced mainly for dissolving polymers with a developed network of hydrogen bonds, including cellulose, N-methylmorpholine-N-oxide (NMMO) [6,7,8], which, as shown in Table 1, has the highest electron-donor activity, measured in relation to the electron-acceptor substances (Table 1). In this characteristic, it surpasses such popular donor solvents as dimethylacetamide (DMAc) and dimethyl sulfoxide (DMSO) [9].

NMMO exists in several thermodynamically equilibrium crystal-hydrate forms, and, with decreasing water content, their dissolving ability increases, but simultaneously the melting point of crystal-hydrates increases [10]. Dissolution of cellulose starts with monohydrate (water content 13.3% and T_m_ ~80 °C). The use of NMMO as a direct solvent of cellulose and its mixtures with polyesters and polyamides is described in numerous publications (see, for example, [6,11,12,13]). As an illustration of the dissolution of cellulose in NMMO, a schematic representation of the interaction stages of the solvent with the polymer is presented in Figure 2. Briefly, NMMO molecules, due to their strong donor properties, first break intermolecular (Stage I) and then intramolecular H-bonds of cellulose Stage II). As a result, the solution is forming.

In the traditional method of fibers or films spinning from hot cellulose solutions in NMMO, the liquid solution jet leaving the spinneret die and passing through the air gap enters a cold-water coagulation bath [14]. As a result of mass transfer processes, upon contact of the solution with the coagulant, phase transformations occur, cellulose loses its solubility and a gel-like fiber is formed, in which the structural features of the future cellulose fiber are laid [15]. However, already in the early stages of coagulation, high crystallization rate of cellulose is observed, which leads to a decrease in the deformation properties of the fibers [16]. It is noted in [17] that the coagulation of the solution at the microlevel is accompanied by the formation of a swollen gel-like composition, and the removal of the liquid phase (solvent plus coagulant) during drying leads to contraction of the fresh-spun fibers. The processes listed above proceed with high speed, and, as a result, already in the coagulation stage, internal stresses appear in the fiber, which impair the mechanical characteristics of the final fibers. To avoid this unwanted situation, we need to control the kinetics of mass transfer at wet and dry–wet jet processes.

Mass transfer mainly concerns inter-diffusion of a solvent and a coagulant. In the case of fast diffusion processes, the surface layers of the gel fiber are rapidly enriched in the polymer, and further diffusion into the depth of the fiber or film slows down. In this case, the so-called “shell–core” morphology forms, which impairs mechanical properties. Such coagulants are called “stiff” and for obvious reasons they are abandoned. In particular, water is a relatively stiff coagulant for cellulose solutions, and it could be “softened” by adding NMMO, i.e., using the dilute solution of a solvent in a coagulant [18]. As a result, the materials spun in such bathes have different morphology and structure compared with materials processed via fast coagulation, which affects the set of mechanical characteristics [19,20]. An increase in the temperature of the coagulation bath containing an aqueous solution of NMMO to 80 °C upon spinning of 8% cellulose solutions leads to a decrease in the thickness of the dense surface shell but with the appearance of a number of defects [21].

Another way to control the kinetics of coagulation in the spinning process from cellulose solutions in NMMO is the use of aliphatic alcohols [22,23,24,25], which are more or less compatible with aqueous NMMO solutions and therefore can be used in the spinning process. It is shown in [22] that, in the case of coagulation of the solution with ethanol, a decrease in the degree of orientation of cellulose macromolecules is observed, which entails a decrease in the elastic modulus and stiffness of the fibers. With an increase in the length of the hydrocarbon fragment of alcohols used as a coagulant, a slowdown in the crystallization of cellulose, a decrease in the hardness and an increase in the elongations at break of the spun fibers are observed [23].

It should be noted that, in the early cycle of works devoted to this issue, a number of scientific and technological approaches were already tested. Thus, it is shown in [24] that, when isopropyl alcohol (IPA) is used as a coagulant, NMMO crystallization is possible in a just-spun fiber. The authors stated that, at coagulation temperature of 20 °C, IPA molecules have a lower affinity for the solvent compared to water. An increase in the IPA temperature (up to 35 °C) and, accordingly, the rate of coagulation leads to the formation of a porous structure with an average pore size of 100 nm and even to the formation of large vacuoles in the fibers [24,25]. When spinning proceeds in IPA of different temperatures, the strength of the fibers falls by almost 2.5 times when the temperature increases from 15 to 50 °C, and the values of elongation at break have a maximum at 20 °C.

In the case of an aqueous coagulant, the problem of solvent regeneration has already been solved, and up to 98% of the used NMMO is returned to the dissolution cycle [26]. It is natural that attempts to use alcohols and their mixtures with NMMO and water as coagulants led to the development of at least draft methods of solvent and coagulant regeneration. Thus, in [27], the phase states of the isobutanol (IBA)–NMMO–water system at different temperatures are considered in detail, which could be the basis of an approach to the development of the main stages of NMMO and alcohol regeneration.

The transition from an aqueous coagulant to a softer one (NMMO aqueous solutions, alcohols, etc.) leads to a radical change in the structure of the formed material [25,28,29], which allows varying the properties of the formed fibers (films). Among these properties, the main attention is devoted to an optimal ratio of mechanical characteristics and resistance to pilling for fibers, porosity and hence selectivity, permeability, etc. for films [21,23,30]. In other words, depending on the ratio of the crystalline and amorphous phases in cellulose, not only the original morphology is formed in the final material, but also a specific system of hydrogen bonds. A priori, these bonds will be weaker in amorphous areas, which leads to fibrillation of one-dimensional products (fibers), but at the same time to an increase in porosity, i.e., permeability, of two-dimensional materials (films) under the influence of a mechanical field [31,32].

An IR spectroscopy study of the NMMO monohydrate–IBA system revealed that the alcohol is bound by hydrogen bonds with water, and so strongly that part of the water can be released from the NMMO and exist as a separate liquid phase or, possibly, a water–alcohol associate, which leads to an interaction between NMMO and alcohol through water [33]. If the cellulose solution is formed in an alcohol bath and then the fiber or film is washed with water, the NMMO and alcohol residues are completely washed out. The system of hydrogen bonds and, as a result, the structure of the material changes drastically, since a less ordered structure is formed than in water [23]. When the fresh-spun material is additionally washed with water, its density increases (due to growth of crystallinity) and even exceeds similar values for samples formed in water [19,29]. In this case, we are talking about a two-stage formation of the material structure by the primary interaction of the solution with alcohol and, finally, with water.

Accumulating the theses presented in the Introduction, we can express certain considerations about the feasibility of a detailed study of two-stage spinning of fibers and films from cellulose solutions in NMMO in water and alcohols of different temperatures. By this approach, we can regulate the stiffness of the coagulant and divide the spinning process into primary coagulation in alcohol and secondary conditioning of fresh-formed materials in water. This article is devoted to these issues.

## 2. Materials and Methods

To prepare cellulose films and fibers, 12% solid-phase mixture based on cellulose (Baikal pulp and paper mill, Baykalsk, Russia) (degree of polymerization ~600, α-cellulose content ~94% and moisture content ~8%,) and NMMO monohydrate produced by Demochem (Shanghai, China) with a water content of 13.3% (T_m_ ~76 °C) were subjected to mechanical activation [34]. During the activation process, the H-complex of NMMO and cellulose formed as a solid solution. The 0.5% Propylgallate (Sigma-Aldrich, Saint Louis, MO, USA) was introduced into the system to suppress the thermal-oxidative destruction. Activated pre-solutions for homogenizing were heated to 100 °C in the operating unit of the Rheoscope 1000 capillary viscometer (Ceast, Torino, Italy) and passed through a capillary with a diameter (d) of 0.5 mm and a length (L) of 10 mm (L/d = 20). As a result, homogeneous dopes were obtained, the quality of which was evaluated using polarizing optical microscope Boetius (VEB Kombinat Nadema GmbH, Ruhla, former DDR). The same viscometer with bath accessory was used for fibers spinning.

To form cellulose films, solutions ejected from a capillary viscometer were collected on polyethylene terephthalate or polyimide antiadhesive films and passed through the rolls of the HLCL-1000 laminator (Cheminstruments, Fairfield, OH, USA) heated to 100 °C. The thickness of the solution layer between the rolls was set using calibration plates, taking into account the presence of antiadhesive films. After preparing an even layer of cellulose solution, the cover film was removed and the sample was immersed on the substrate in either water or isopropyl or isobutyl alcohols (Component-Reagent, Moscow, Russia) at two temperatures—ambient and 80–90 °C. The volume of the bath was 1 L. The slide antiadhesive film was removed in a coagulation bath at the stage of gel-like film formation. Coagulation of the solution layer was proceeded during the day. Then, the films were washed with either alcohol or water. The washing liquids were replaced once a day. The washing procedure was repeated at least twice. Films were dried at room conditions on the frame (isometric conditions) to an equilibrium moisture content (~8%).

The viscosity of NMMO solutions in IPA or in IBA with different component ratio was measured on a rotational rheometer Physica MCR 301 (Anton Paar, Graz, Austria) in a cooling mode from 100 to 20 °C. The plate-plate operating unit (50 mm diameter) was used for measurements. The tests were performed under steady flow conditions at constant shear rate (1 s^−1^).

The compatibility of the NMMO–coagulant system, as well as the phase state of the cellulose solution in NMMO when coagulated, was studied by optical interferometry. The method of conducting the experiment did not differ from the traditional one [35,36]. The components were brought into contact in the “side-by-side” version in a cell of a wedge-shaped gap formed with semi-reflecting glasses. Previously, their refractive indices of reagents were measured. In the case of transparent components, two sets of interference fringes separated by a phase boundary are registered in monochromatic light. The interference patterns pitch is inversely proportional to the refractive index of the given liquid. During interdiffusion, the interference bands near the phase boundary are curved and the intensity of diffusion is determined by the kinetics of changing the shape of the bands. When the components are fully compatible, the phase boundary disappears; when the interference patterns are partially compatible, the analysis of the obtained interference patterns allows us to evaluate the concentration profiles of components in the diffusion zone and construct the phase diagrams [37]. In the case of the NMMO–coagulant pair, measurements were carried out in step isothermal mode, gradually increasing the temperature, in the range of 25–90 °C at a coagulant temperature of 25 and 80–90 °C.

The thermal analysis device DSC 823 e (Mettler Toledo, Greifensee, Switzerland) was used to evaluate the thermal behavior of the NMMO–coagulant system. Measurements were made in sealed aluminum crucibles with a volume of 40 µL at a temperature change rate of 10 °C/min. Temperature program of the experiment was: first heating 25–80 °C; isotherm 80 °C, 5 min; cooling 80 to 50 °C; isotherm 50 °C, 5 min; and second heating 50–80 °C. The consumption of inert gas (argon) was 70 mL/min.

Simultaneously with optical interferometry and DSC methods, the turbidity method was used to observe appearance of crystalline phase. The transition points were recorded visually and using an optical microscope “Boetius” equipped with a heating table.

The structure of the films was studied by X-ray diffractometry on the Rigaku Rotaflex D/MAX-RC (Rigaku Corporation, Tokyo, Japan) installation equipped with a rotating copper anode (operating mode of the X-ray source is 50 kV, 100 mA, graphite monochromator on the beam reflected from the sample, the wavelength of the characteristic radiation λ = 0.1542 nm), a horizontal goniometer and a scintillation detector. X-ray survey was conducted in geometries “reflection” and “transmission” under the scheme Bragg–Brentano in the continuous θ–2θ scanning in the angular range 5–30° with a scan step of 0.04° at room temperature.

For registration IR spectra of NMMO after combining with IBA and water and their evaporation IR microscope HYPERION-2000, connected with IR Fourier spectrometer IFS-66 v/s Bruker (crystal–Ge, scan 50, resolution 2 sm^−1^, frequency range 4000–600 sm^−1^). The morphology of the surface and transverse cleavages of dried cellulose films was studied by low-voltage scanning electron microscopy (SEM) on a FEI Scios microscope (Hillsboro, OR, USA) at an accelerating voltage of less than 1 kV in the secondary electron mode. The amount of residual solvent in the films was determined using a multi-element Flash 2000 CHNS/O analyzer (Thermo Fisher Scientific, Paisley, UK).

## 3. Results and Discussion

NMMO monohydrate is a white crystalline substance. When it is added to alcohol, you can visually track how the appearance of the system changes under room conditions (see Figure 3 for the example of IPA).

If a system with an equal content of components is a single-phase liquid, then, with a further increase in the concentration of the solvent in alcohol, the phase separation takes place. When the NMMO concentration reaches 80%, the mixture becomes a paste, and, with a component ratio of 10:90, the NMMO crystals look like wet powder. It is possible to imagine that similar changes in the consistency of the system occur at spinning films (fibers) from cellulose solutions in NMMO, unless, of course, the presence of the polymer component is taken into account.

The change in the consistency of the binary solvent–coagulant system on a temperature scale was estimated using rotational rheometry for NMMO solutions in IPA when they were cooled from 100 °C to room temperature (Figure 4).

Temperature scanning is an express method that allows detecting the behavior of the system during cooling and determine the points at which there is an extreme jump in the viscosity values. For the studied solutions, with an increase in the alcohol content, there is a shift in the flow loss temperature to the low temperature region. Of course, the detected points are not absolute, but, as well as visual observations of the appearance of the system at room temperature, allow us to determine the approximate points of transition to the crystalline state.

To detect more correctly the melting temperatures of NMMO solutions in IPA, they were studied using the DSC method (Figure 5).

As is seen from the DSC curves, the maximum endo effect corresponding to the melting of IPA is observed at a temperature of -89.8 °C. According to [38], at this melting point, the water content in alcohol does not exceed 2%. On the other hand, the melting point of NMMO monohydrate (13.3% water content) is 76 °C. When alcohol is added to NMMO, the maximum of the endo peak corresponding to the NMMO monohydrate melting is shifted towards low temperatures Thus, at 13% IPA, the melting point of the solvent is 61 °C; at 28%, it falls to 42 °C; and, at an alcohol concentration of 42%, the melting occurs almost at room temperature (24 °C). These ratios of temperature and composition of the binary mixture form the liquidus line. Additional endo peaks in the area of 14–15 °C can be associated with the redistribution of water in the system and the appearance of NMMO hydrates with a higher water content. The presence of exo peaks on Curves 3 and 4 seems to be due to the crystallization of NMMO monohydrate diluted with alcohol, i.e., an appearance of crystal solvates NMMO–IPA.

Visualization of interference fringes at the diffusion interaction of two substances is possible only if they are transparent in visible light. Up to the melting point, NMMO is crystalline, and therefore opaque. However, the second substance—Isopropanol (or Isobutanol)—is transparent under these conditions and generates interference bands. In such cases, the interaction of components can be judged by half of the total interference pattern related to alcohols.

Figure 6 shows interferograms of the interaction zone of the NMMO–IPA components. The dark region observed on interferograms is a crystalline opaque phase of NMMO (Figure 6a,b), so up to the melting point of the solvent, the interaction of substances is considered only by one half of the interference pattern related to the transparent IPA. At a temperature of 30 °C, a curvature of the interference bands of the IPA in the contact zone is observed, indicating the diffusion of crystalline NMMO into alcohol. This indicates partial compatibility of components under these conditions. At T > T_m_ of NMMO, the interferogram shows a region of monotonic transition of the IPA interference bands to the corresponding bands of transparent NMMO melt and the absence of a phase boundary (Figure 6c), i.e., the components are fully compatible.

Interference patterns give information about the concentration profiles of components, and, based on the composition of the terminal zones, construct a phase diagram (Figure 7). The diagram also shows data from the DSC and the turbidity point method.

The system under study is characterized by a crystalline equilibrium. In Area I, the components are fully compatible (single-phase state), while, in Area II, the phase separation proceeds. The close to each other liquidus lines, obtained by various methods, evidence their reliability.

The phase state diagram gives possibility to select conditions for obtaining films that exclude possible crystallization of the solvent during coagulation. Photos of the obtained films coagulated and kept for two days in IPA, as well as films washed after such treatment in water at room temperature and dried, are shown in Figure 8.

As is seen in the photos, the film washed in IPA is turbid and only after holding such a film in water and drying it becomes transparent. Changes in the optical properties of the film can be associated with both structural and morphological features [19,39,40,41] and the presence of a residual solvent [33]. The study of the chemical composition by elemental analysis showed that the nitrogen content in the film formed and washed in IPA is about 0.5%. This value indicates a significant fraction of the solvent in the cellulose, which, apparently, affects the appearance of the film.

### 3.1. Low-Temperature Coagulation

The most important part of the research is modeling the fibers and films spinning into various coagulants. For this purpose, the interaction between a dope of cellulose and isopropyl alcohol at different temperatures of the latter was studied by optical interferometry. The corresponding interferograms at the alcohol temperature of 25 °C are shown in Figure 9.

Within the first minute after the contact of the cellulose solution in NMMO with the IPA, the darkened zone that characterizes the coagulation of the solution and forming film is about 100 microns, i.e., the process is quite fast. In this area, the formation of defects in the form of droplets–vacuoles (marked with arrow in Figure 9b) is observed. The same defects were described earlier in a study on coagulation of a drop of cellulose solution with isopropanol [25]. Vacuoles grow in the direction of the diffusion front, and their maximum size reaches 20–30 microns. Judging by their optical contrast with the deposited film, the vacuoles are filled with a liquid phase.

As the coagulation front moves away from the phase boundary, its propagation speed decreases and after 5 min the width of the coagulation area is ~150 microns. The decrease in the process rate is understandable, since the coagulant for contact with a fresh portion of the solution must diffuse through the already coagulated part of the film. In the resulting films, despite the observed coagulation, the solvent remains in significant quantities.

Along with this work, the process of coagulation of a cellulose solution into isobutanol was studied in [33]. If we compare the rates of coagulation of the solution with isopropanol and isobutanol, the coagulation process is almost twice as slow for isobutanol. After 45 min, the position of the dark zone corresponding to the formed film practically does not change its position, i.e., the interaction is “frozen”. The intensity of the formation of a liquid defective phase (vacuoles) and their number for IBA, on the contrary, is significantly higher.

Thus, the release of the polymer phase occurs in contact with IPA much faster than with IBA, and the number of drops of the liquid phase becomes much smaller. Apparently, it is the formation of liquid droplets by coagulation of a solution of cellulose in IPA and IBA is a source of major defects in the dried film. As mentioned above, this defect is mainly inherent in films obtained by coagulation of the solution in IBA [33]. As an example, a microphotograph of the transverse cleavage of such a film is shown in Figure 10a.

Films formed in IPA have a less defective morphology (Figure 10b). Visually, the morphology of the film cross-section consists of three zones: the upper one with a dense morphology that contacts the alcohol during coagulation; the defective one, whose thickness reaches 10 microns; and the lower one with a uniform texture of the same thickness that contacts the substrate. The upper surface, which had a direct contact with alcohol, due to the contraction of large defects during drying, has a wavy shape. The lower surface remains flat.

The most rapid process of coagulation of the cellulose solution occurs when it is spinning in water. In this case, after just 1 min, the length of the film formation zone has reached its thickness, while corresponding duration of inter-diffusion for the IPA is of the order of 20–30 min. Scanning electron microscopy image of the film cross-section spun in water at 25 °C is presented in Figure 10c. Thus, water is the most rigid precipitant, the action of which consists in a rapid dilution of the solution and the loss of aqueous NMMO solubility in relation to cellulose. It is possible that the mechanism of action of alcoholic coagulants is different, and these features are discussed below.

One of the most important indicators that reflects the coagulation mechanism is the structure of films at different stages of preparation. Coagulation of cellulose films in a “softer” IPA leads to the formation of a different structure than when forming into water. X-ray diffractograms of films formed in IPA at room temperature, washed in alcohol or water and dried in room conditions are shown in Figure 11.

The diffractogram of regenerated cellulose II is characterized by the presence of maxima with angular positions 2θ ~12.3°, ~20.5° and ~21.9° [42]. The diffractogram of the film sustained in water (Figure 11, curve d) shows a peak with an angular position of 2θ ~12° and a wide peak in the region of ~20°. They correspond to reflections from the crystallographic planes $\bar{1}\bar{1}$0, 110 and 020, while the peak at 2θ ~12° is less intense compared to the peak of 2θ ~20.5°.

On the diffractogram of the film formed into alcohol obtained in the reflection mode, there is only one wide peak in the region of 2θ ~20°, which corresponds to the amorphous cellulose [43,44,45,46]. From this, it can be concluded that, in fresh-formed films obtained in IPA, it does not allow completely removing NMMO from the sample and leads to the formation of a chaotic structure, which is characterized by a complex set of interactions between cellulose, alcohol, solvent and water. We assume that they contain areas where complete regeneration of cellulose with the structure of cellulose II has occurred and prevailing areas of amorphous cellulose, both of which contain a solvent [47].

This seems to be a general pattern for cellulose films and fibers formed in alcohols. It is noted in [29] that the fibers formed from cellulose solutions in NMMO to ethanol also contain a solvent. To remove solvent residues from films and fibers formed in IPA, additional washing with water of just-spun precursors is necessary, followed by drying. With this processing, the intensity of reflexes $\bar{1}\bar{1}$0 and 110 is redistributed (Figure 11, curve d). The intensity of the first reflex increases, and the second decreases, which indicates the change in the system of hydrogen bonds and transition a part of the amorphous cellulose to crystalline state.

### 3.2. High-Temperature Coagulation

As stated above, the melting point of NMMO monohydrate exceeds 75 °C. Under the conditions described above, the spinning was carried out in cold water or alcohols. This does not preclude partial crystallization of the solvent in dry–wet jet and wet-spun fibers and films. Increasing the temperature of the coagulant to the level eliminates this effect, as well as accelerates the diffusion processes that lead to the release of the polymer phase from solutions. It can be expected that the structure of the materials prepared in such a way will be specific. For this purpose, the interdiffusion processes was organized at almost the same temperature as the dope (12% solution of cellulose at 80 °C). For this aim, the coagulants were heated to the same temperature. The interferograms reflecting the mass transfer kinetics between the dope and IPA are shown in Figure 12.

As can be seen from the interferograms, the interdiffusion between two phases takes place, since the curvature of the bands occurs both in the solution phase and in the alcohol phase, i.e., on both sides of the interface.

The forming film is opaque and therefore looks dark on microphotograph. Over time, it expands towards the cellulose solution. The process of coagulation of the solution into a “hot” alcohol is characterized by a higher rate of mutual diffusion, as evidenced by the rate of film growth (Figure 12) and the rate of diffusion front propagation (Figure 13). At 1 min from the beginning of contact in the case of “hot” coagulation in IPA, ~0.15 mm of film is formed, while, in the case of “cold” one, only 0.1 mm.

For comparison, the same graph shows the coagulation profiles of the same solution in IBA and water at two temperatures. The rate of the diffusion process and appearance of the just-formed film is most clearly reflected by the position of the coagulation front at 1 min after start of contact. These data are shown in Table 2.

It is important that the rate of cellulose film formation is the same for IBA at 90 °C and IPA at 25 °C, which emphasizes the stronger coagulating effect of IPA. Nevertheless, water is the most active coagulant, especially at high temperature. Thus, the range of activity of coagulants for cellulose solutions is as follows: IBA < IPA < water. IPA and water are compatible at any composition, but, for IBA–water pair, the compatibility is not complete and amorphous phase equilibrium described by binodal exists [27]. At room temperature, the one-phase solutions forms at IBS content below ~12% and above ~85% only.

To understand the structural changes in cellulose caused by an increase in the temperature of the coagulant, the diffractograms shown in Figure 14 were taken.

For films obtained through alcohol of different temperatures and kept in IPA or water, the type of diffractograms is the same (see Figure 11; Figure 14). Comparison of the ratio of integral intensities of peaks 2θ ~12° and ~20.5° corresponding to the crystallographic planes $\bar{1}\bar{1}$0 and 110 (transmission mode) for films formed into alcohol of different temperatures, washed with IPA and dried showed that, in the case of “cold” alcohol, it is less than in the case of “hot” coagulation. Changes in the intensities of these peaks reflect structural changes in cellulose, but the transition from alcohol to water washing, regardless of temperature, diminishes the differences in the ratio of intensity of these maxima.

### 3.3. About Mechanism of Coagulation

Deposition of the polymer from the solution under the action of coagulants is a complex multi-factorial process. According to [48], when in contact of the solution jet and coagulant, the most probable mechanism of the solution phase decomposition is changing interaction of polymer and solvent, described by the Flory–Huggins parameter χ, by diluting the solution with coagulant. Using the phase diagram for amorphous phase equilibrium and assuming that, according to the classical lattice model, χ is proportional to the reciprocal temperature, in the experimentally observable variables “T–C”, this phase diagram is presented in Figure 15 (for simplicity, a region of metastable state, described by spinodal, is not shown).

The mass exchange (diffusion interaction) changes χ, which leads to drastic changes in the phase equilibrium in the system, namely to the separation into two coexisting equilibrium phases of different composition. Bearing in mind all the conditionality of this thermodynamic approach, for which only the initial and final states are important, we note that, in the case of a cold coagulant, the content of the polymer C_2_ in the concentrated phase is higher than in the case of isothermal (hot) spinning (C**). Hence, we should expect a denser structure of the gel-fiber in the first case and a looser one in the second. Similar processes occur in the case of crystalline equilibrium and in the superposition of amorphous and crystalline equilibria [49,50]. In the latter case, the formation of a crystal-solvate phase is possible.

It should be noted that this thermodynamic approach works mainly for athermic solutions formed as a result of solvation of macromolecules, i.e., due to Van der Waals (dispersion) forces. In the case of cellulose solutions in a strong donor–acceptor solvent, the components of the solution are connected by stronger interactions, including H-bonds. For such a system, it is necessary to take into account not only the kinetics of the coagulation process but also its mechanism. From the point of view of kinetics, we already know that water is the stiffest and IBA the softest coagulant. According to existing concepts [51], a rigid coagulant quickly forms a dense gel-like surface film (shell), through which diffusion is difficult, and internal cavities of the fiber/film can maintain the solution for a long time. When the shell breaks due to occasionally uneven coagulation, large defects are formed, and it is for such coagulants that a defective cross-section of fibers or films of the “shell–core” type is formed. In the case of soft coagulants, the polymer deposition process is longer, but also more uniform, although the initially given loose morphology can be preserved in the finished material.

However, general considerations are not always able to describe reality. Thus, the most uniform morphology of the films is obtained in the case of water, although it is the most rigid coagulant for these solutions compared to alcohols. This may be due to the scale factor, i.e., the thickness of the coagulated part of the fiber or film. For thin fibers and films, the proportion of the hard shell can be comparable to the thickness of the sample. Another reason may be a sufficiently permeable surface film that does not interfere with diffusion processes, especially in the case of mutual diffusion. However, in any case, the role of the nature and intensity of intermolecular interactions in a complex multicomponent system cannot be excluded.

With respect to intermolecular interactions, the most interesting are the connections of isomeric alcohols with NMMO, since these components are primarily involved in the process of interdiffusion, when the spinning solution jets come into contact with the coagulant. For their analysis, the IR spectra of NMMO obtained from solutions in the softest coagulant (IBA) and in the hardest (water), after evaporation of volatile components: alcohol or excess water, were taken and analyzed (Figure 16).

When analyzing the spectra, attention is drawn to the very high intensity in the domain of 3400–3500 cm^−1^ (-OH groups) and at 1660 cm^−1^ (band of H-O-H strain oscillations) in the NMMO spectrum from the IBA. In addition, the presence of a 675 cm^−1^ band in this spectrum reflects vibrational oscillations of water. Both bands belong exclusively to the vibrations of water molecules, and they are more intense in the NMMO spectrum from IBA than in the NMMO spectrum from water. It follows that, in the NMMO–water–IBA system, alcohol is connected by hydrogen bonds with water, and so strongly that part of the water is released from the NMMO crystal hydrate and exists as a separate liquid phase or, possibly, as a water–alcohol associate. The existence of such associates is indicated by a wide band at 2250 cm^−1^. Therefore, it can be argued that the interaction of NMMO crystal hydrates with IBA passes through water.

Intensive diffusion of alcohols with a high affinity for water initiates the process of water redistribution in the system and, accordingly, saturation of the diffused alcohol stream with water. Drops of water solution in alcohol appear in the contact area (see interferogram in Figure 9). Judging by the bending of the interference bands, diffusion is active on both sides of the phase boundary, but the question of what diffuses from the solution side requires additional reasoning. As mentioned above, the strong interaction of alcohol with water leads to the separation of crystallization water from the NMMO molecules and its interaction with alcohol. The electron-donor activity of NMMO increases in this case, as free unshared electron pairs appear on the oxygen atom of N-O groups, and NMMO molecules start to interact with the hydroxyl groups of alcohol. In other words, the NMMO molecules replace the crystallization water with alcohol molecules, which leads to a radical change in the state of the system as a whole. Instead of the NMMO monohydrate molecule, which is a cellulose solvent, alcohol–NMMO solvate is formed, which is a thermodynamically poor solvent (rather, a precipitant) of cellulose.

Knowledge of the phase equilibrium in the studied solvent–coagulant and solution–coagulant systems allows us to assert that the deposition of cellulose from the solution in NMMO at contact with alcohols proceeds in two stages. At the first stage, as a result of selective interaction of alcohol with water presented in NMMO monohydrate, an amorphous film of actually solid solution is formed, which only after prolonged exposure in water transforms into a swollen film of hydrate-cellulose with its inherent amorphous-crystalline structure.

The temperature of the coagulant has a significant influence on cellulose film/fiber morphology. In the case of a “hot” coagulant, a more defective structure is formed due to the appearance of “strands” (elongated voids–vacuoles) in the direction of the alcohol stream (Figure 17).

Microphotographs of the cross sections of cellulose films formed in a “hot” coagulant fully reflect the processes occurring during the diffusion interaction of components detected by interferometry.

Large vacuoles after washing in alcohol (Figure 17a) originate near a thin surface layer, and their height can exceed 45 μm, with the thickness of 10–15 μm. These cavities are separated by partitions (~5 microns) with a dense shell and a mesh core with an average pore size of 500 nm. The lower part of the film has a thickness of up to 8 microns and is characterized by a dense texture. This texture corresponds to the morphology of fibers/films obtained in a stiff coagulant. When the film after coagulation in alcohol is further washed in water and dried, the resulting vacuoles partially collapse, which leads to the formation of a wavy surface of the film. Shrinkage during drying results in a decrease in the number of visible vacuoles and pores and the formation of depressions on the surface of the film.

## 4. Conclusions

The role of various polar coagulants in the deposition of cellulose from the solution in NMMO and the formation of a certain morphology and structure of regenerated cellulose was studied. From the point of view of the efficiency of solution coagulation, the studied coagulants are located in the series: H_2_O > IPA > IBA. When using alcohols, the process of forming a cellulose fiber/film proceeds in two stages: in the first stage, the alcohol/NMMO solvate is formed, and the amorphous cellulose associated with alcohol and residual NMMO is released. In the second stage, as a result of long-term processing of this associate with water and drying the hydrate cellulose (HC) forms. The very rough scheme of the solution coagulation and deposition of cellulose is shown in the diagram (Figure 18).

In the left part of the diagram, an additive compound–IPA/NMMO solvate and cellulose with a glassy solvent are formed. In the right part of the diagram, the formation of hydrate cellulose occurs as a result of washing the multicomponent system prepared at the first stage with water. The most favorable morphology for engineering materials is realized when water is used as a coagulant. Application of IPA and IBA leads to the formation of a loose morphology with a large number of pores of different sizes that can be controlled. Such films can serve as precursors of membrane materials for the separation of gases and liquids.

## Figures and Tables

**Figure 1 materials-13-03495-f001:**
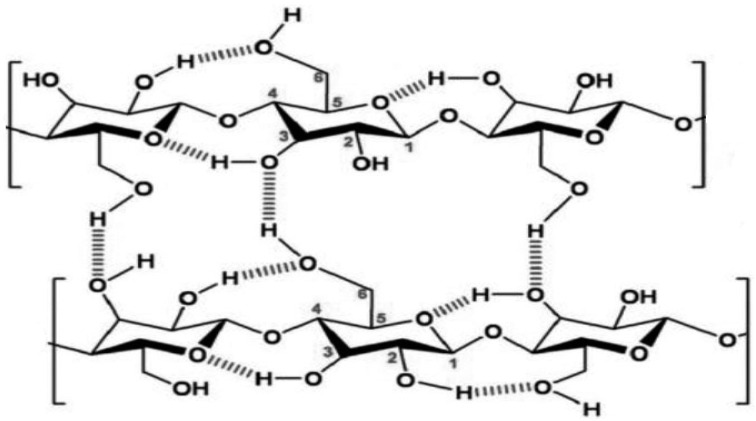
Structural formula of cellulose.

**Figure 2 materials-13-03495-f002:**
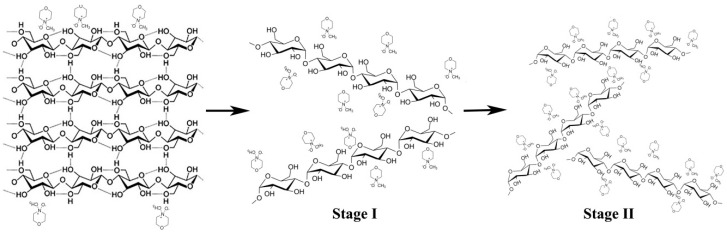
Schematic representation of the stages of interaction of NMMO with cellulose.

**Figure 3 materials-13-03495-f003:**
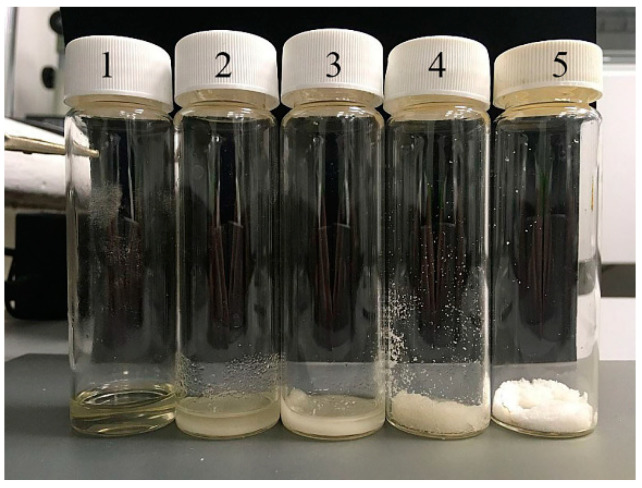
Photos of NMMO–IPA mixtures at room temperature with NMMO content: (1) 50; (2) 60; (3) 70; (4) 80; and (5) 90 (mass %).

**Figure 4 materials-13-03495-f004:**
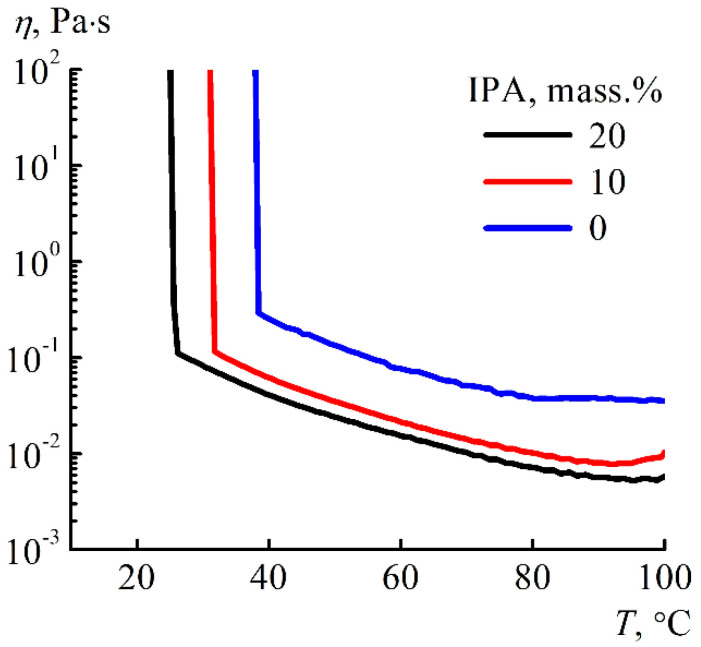
Temperature dependence of the viscosity of NMMO monohydrate with different alcohol content.

**Figure 5 materials-13-03495-f005:**
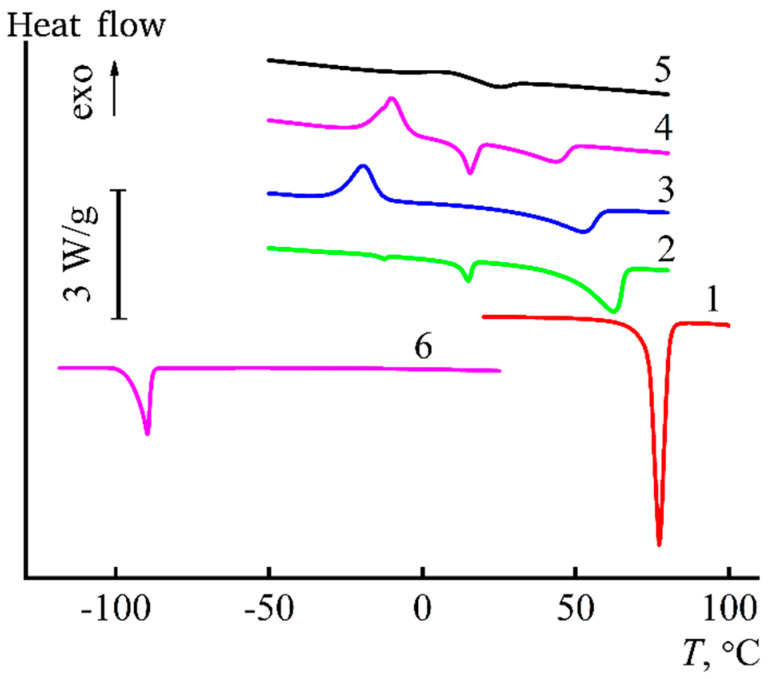
Thermograms of the solvent-coagulant mixtures of the following NMMO/IPA (mass %) composition: (1) 100/0; (2) 87/13; (3) 81/19; (4) 72/28; (5) 58/42; and (6) 0/100.

**Figure 6 materials-13-03495-f006:**
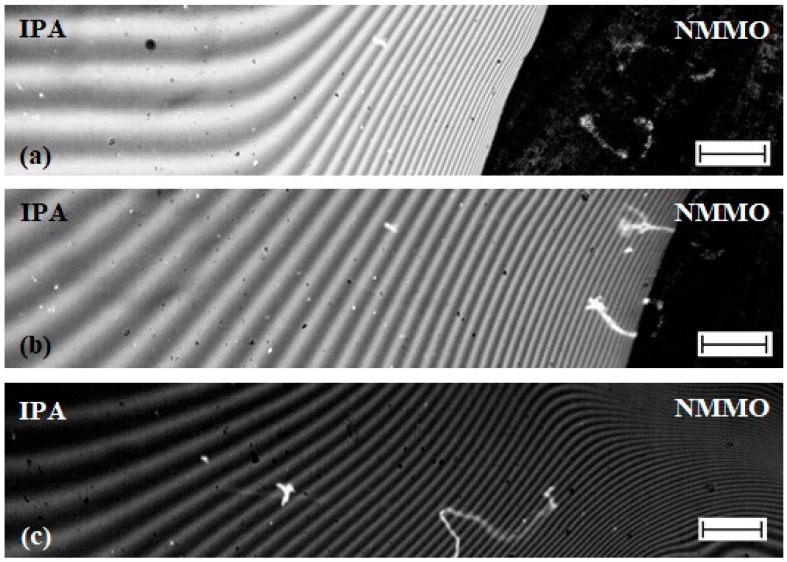
Interferograms of the interaction zone of the NMMO–isopropanol system at temperatures: of 30 85 °C (**a**); 60 85 °C (**b**); and 85 °C (**c**). The length of the scale bar is 250 microns.

**Figure 7 materials-13-03495-f007:**
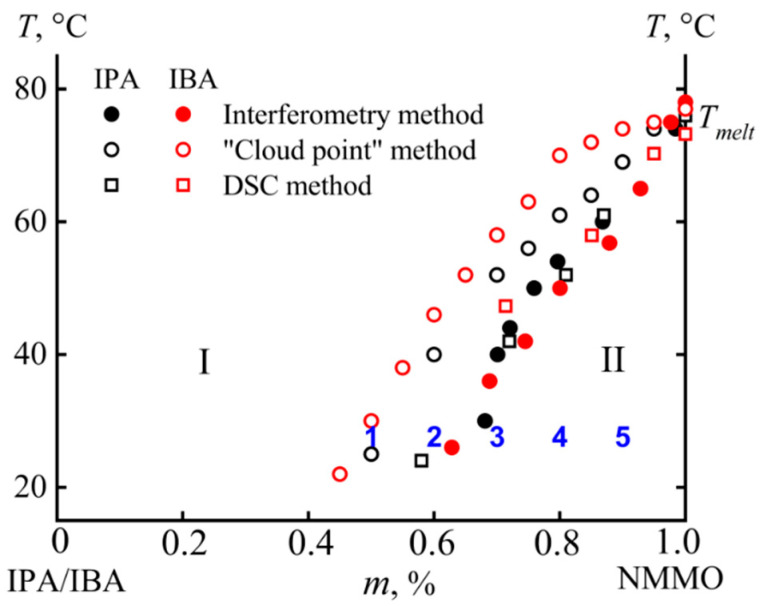
Phase diagram for the NMMO–alcohol system: (I) compatibility area; and (II) phase separation area. The numbers in the diagram correspond to the samples shown in Figure 3.

**Figure 8 materials-13-03495-f008:**
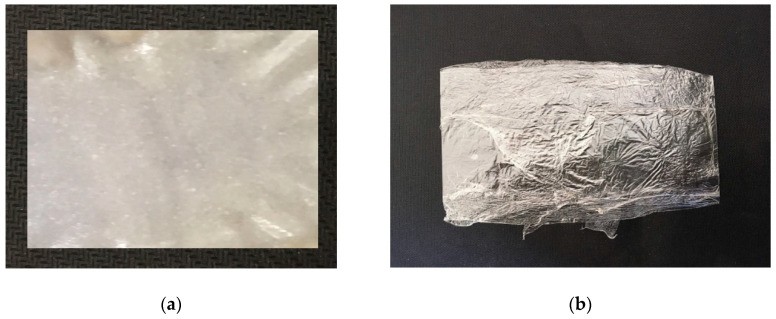
Photos of cellulose films obtained by coagulation of solution in IPA: aged in isopropanol and water (**a**); and dried (**b**).

**Figure 9 materials-13-03495-f009:**
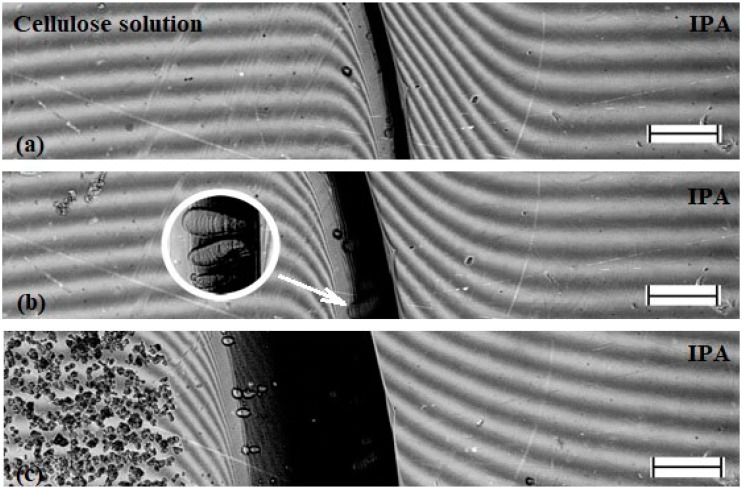
Interferograms of the contact zone of cellulose solution in NMMO and IPA during “cold” coagulation. The time of the experiment: (**a**) 1 min; (**b**) 5 min; and (**c**) 1 h. The length of the scale bar is 250 μm.

**Figure 10 materials-13-03495-f010:**
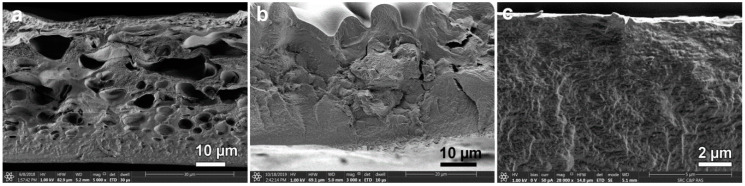
SEM microphotographs of the cross-section of dry cellulose film: spun in IBA at 25 °C (**a**); after coagulation in IPA at 25 °C and further washed in water (**b**); and after coagulation in water at 25 °C, washing in water and drying (**c**).

**Figure 11 materials-13-03495-f011:**
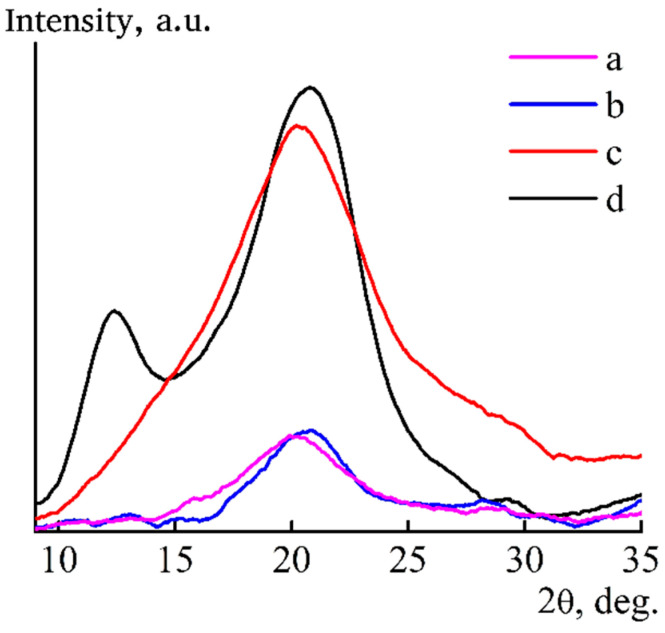
X-ray diffractograms of cellulose films formed in IPA (T = 25 °C) and then aged in alcohol (a,c) or water (b,d) and dried at room conditions in transmission (a,b) and reflection (c,d) modes.

**Figure 12 materials-13-03495-f012:**
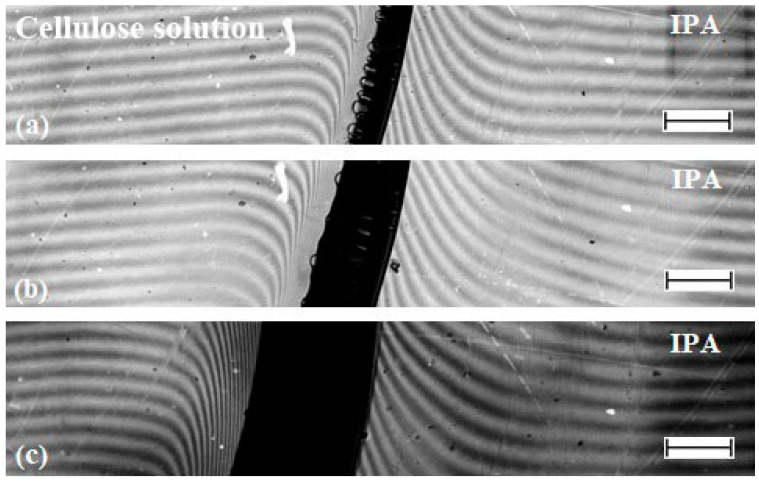
Interferograms of the contact zone of cellulose solution-isopropanol at a temperature of 90 °C. Time from the moment of reagents contact: (**a**) 1 min; (**b**) 3 min; and (**b**)-10 min. The scale bar length is 250 μm.

**Figure 13 materials-13-03495-f013:**
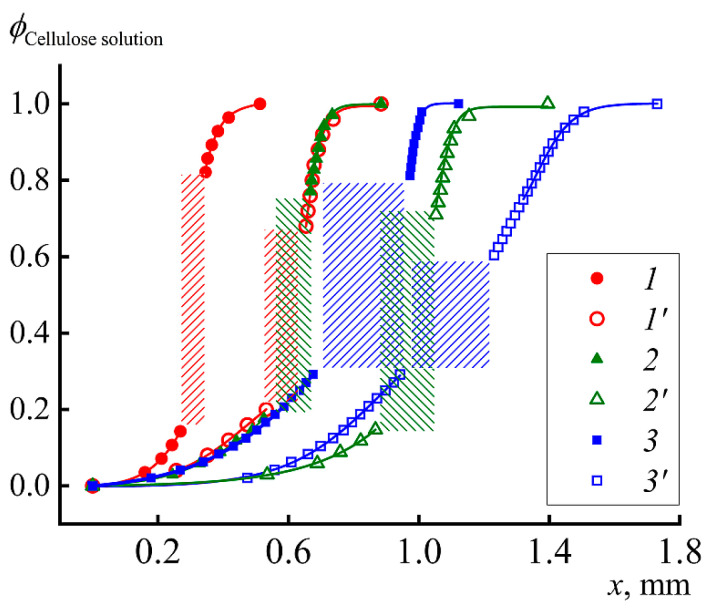
Position of the coagulation front in 1 min after moment of reagents contact (the shaded domains). Coagulants: IBA 25 °C (1); IBA 90 °C (1’); IPA 25 °C (2); IPA 80 °C (2’); H_2_O 25 °C (3); and H_2_O 90 °C (3’).

**Figure 14 materials-13-03495-f014:**
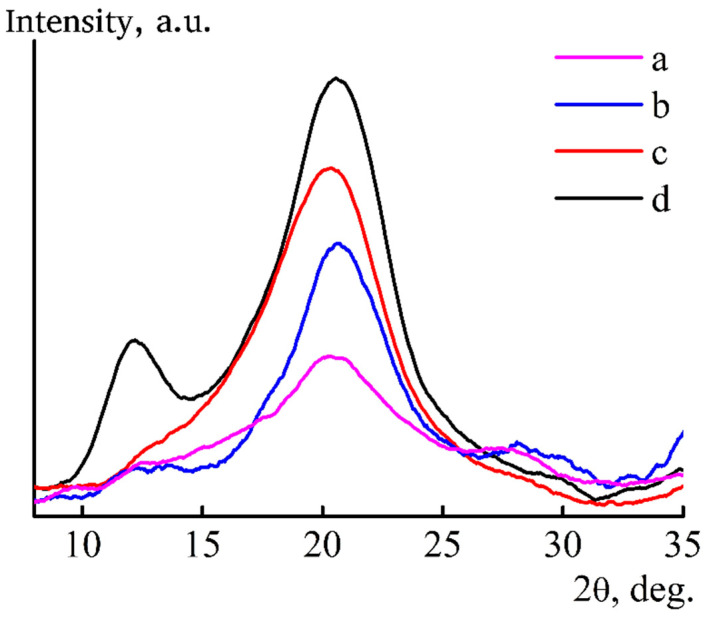
X-ray diffractograms of cellulose films formed in IPA (T = 80 °C) and then aged in alcohol (a,c) or water (b,d) and dried at room conditions in transmission (a,b) and reflection (c,d) modes.

**Figure 15 materials-13-03495-f015:**
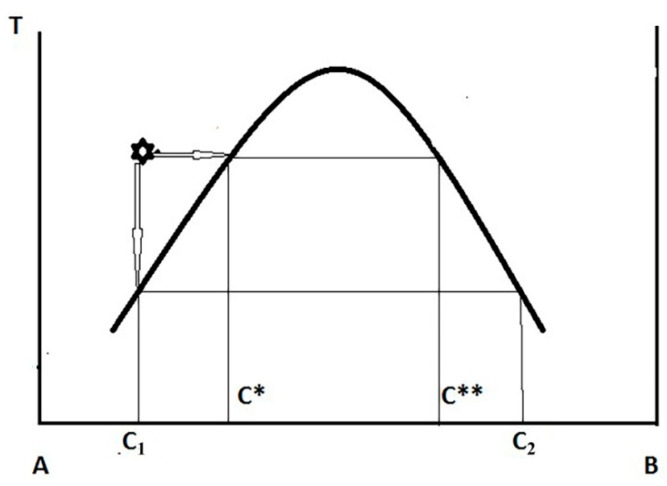
Schematic diagram describing the phase equilibrium in the solvent (**A**)–polymer (**B**) system, and the composition of coexisting phases during “cold” and “hot” spinning a cellulose solution. An asterisk indicates the initial position of the dope.

**Figure 16 materials-13-03495-f016:**
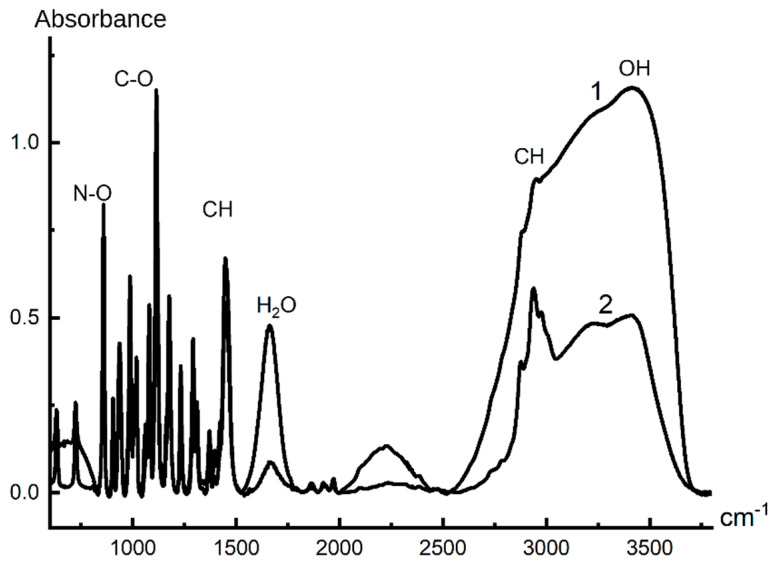
IR spectra of NMMO obtained from: IBA (1); and water (2).

**Figure 17 materials-13-03495-f017:**
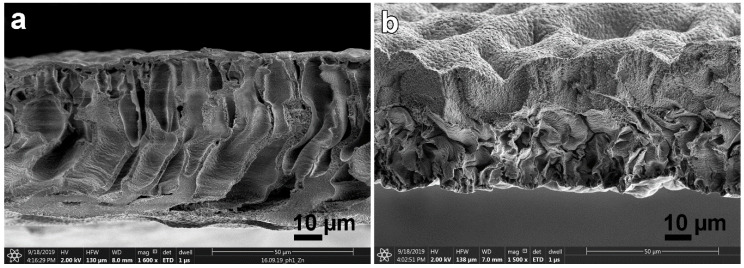
SEM microphotographs of the cross section of dried cellulose films, after coagulation in the IPA at temperature of 80 °C: washing with alcohol and dried (**a**); and washing with water and dried (**b**).

**Figure 18 materials-13-03495-f018:**
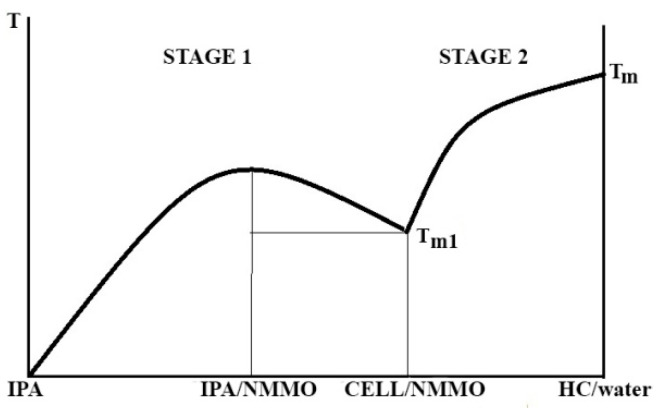
Schematic diagram of the formation of a cellulose fiber/film at interaction of a cellulose solution with isopropanol (amorphous equilibrium) and deposition of hydrate cellulose by water (crystalline equilibrium).

**Table 1 materials-13-03495-t001:** Complexation constant K of donor solvents with phenol and cellobiose in chloroform.

Solvent	Electron-Donor Group	K (L/mole) at 20 °C	
		Phenol	Cellobiose
NMMO	N-O	0.89	0.51
Dimethylacetamide (DMAc)	C = O	0.71	-
Dimethylsulfoxide (DMSO)	S = O	0.47	0.4

**Table 2 materials-13-03495-t002:** Positions of the coagulation front (A) at 1 min after contacting for different coagulants at various temperatures.

Coagulant	IBA (25 °C)	IBA (90 °C)	IPA (25 °C)	IPA (80 °C)	H_2_O (25 °C)	H_2_O (90 °C)
A, mm	0.3	0.6	0.65	0.9	0.8	1.1

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
