# Peer review of "A Role of Coagulant in Structure Formation of Fibers and Films Spun from Cellulose Solutions"

_materials, 2020, doi:10.3390/ma13163495_

Round 1

Reviewer 1 Report

In this work, the authors studied the role of different polar coagulants in the deposition of cellulose from NMMO solution. The resulting morphology and structure of the regenerated cellulose was studied. This is a nice comprehensive work for this topic and the manuscript is well written. I only have one minor comments about the DSC thermograms in Figure 5. Is it possible to correlate the heat calculated from the curves with the explanations in the discussion?

Author Response

Dear Reviewer, we appreciate your detail analysis of this manuscript.

Comments and Suggestions for Authors

In this work, the authors studied the role of different polar coagulants in the deposition of cellulose from NMMO solution. The resulting morphology and structure of the regenerated cellulose was studied. This is a nice comprehensive work for this topic and the manuscript is well written. I only have one minor comments about the DSC thermograms in Figure 5. Is it possible to correlate the heat calculated from the curves with the explanations in the discussion?

Response:

NMMO due to high donor activity forms two equilibrium crystalhydrates (monohydrate and 2.5- hydrate) and several non-equilibrium sesquihydrates. At interaction of NMMO monohydrate with isopropanol a redistribution of water proceeds and formation not only sesquihydrates, but also NMMO-IPA crystal solvates takes place. That is why at increasing IPA content the additional endo- and exo- peaks appear, but to recognize their nature impossible. This is the main reason why we did not analyze values of heats, but consider mainly positions of phase transitions.   

Reviewer 2 Report

Dear Authors,

Presentation of your results on the role of coagulant in structure formation of fibers and films spun from cellulose solutions could be improved as follows:

  • describe IR measurements in Materials and methods section.
  • Figures 10, 11 and 12 could be merged into a single Figure.
  • Fig. 17 does not have scale numbers. Please include phases in the Fig 17
  • Try to extract more information on the H bonding pattern from IR data.

Author Response

Dear Reviewer, thank you very much for your detailed analysis of this manuscript. All your notes are accepted.

Presentation of your results on the role of coagulant in structure formation of fibers and films spun from cellulose solutions could be improved as follows:

  • describe IR measurements in Materials and methods section. Done!
  • Figures 10, 11 and 12 could be merged into a single Figure. Done!
  • Fig. 17 does not have scale numbers. Please include phases in the Fig 17. This is schematic diagram describing amorphous phase equilibrium for two-component system.  Above binodal the homogeneous solution exists. Under binodal the phase separation takes place leading to co-existence of two solutions of different concentration: C1 and C2 or C* and C**. Content of the each phase, i.e. scale factor, can be determined according to the so-called “lever rule”. But for the aim of this paper the most important is difference in the phase concentration prepared in isothermal and “cool” spinning conditions.  
  • Try to extract more information on the H bonding pattern from IR data. Partially, corrected.

Reviewer 3 Report

Comment for materials-884273 is listed in the attachment.

Author Response

Dear Reviewer, we appreciate your detail consideration of our manuscript. Thanks a lot!

Comment for materials-884273 is listed as follows,

There are some miss been named and error typing. We tried to correct.  

(1) In the Abstract, please change “SEM” into “scanning electron microscopy (SEM)”.

(2) In the Keywords, please check "films formation", it did not be found in the manuscripts. Films formation are mentioned in lines 157-168.

(3) In line 55, please define the abbreviations for “DMAA” and “DMSO”. Done!

(4) In lines 62-63, lines 66-67, lines 206-207 and lines 233-234, they are very short not enough to become a paragraph. Done!

(5) In line 127, please define the abbreviations for “IR”. I do not see necessity to describe this well-known abbreviation as infrared spectroscopy.

(6) In line 214 and in the Figure 4, please check "." in the "mass.%". This is usual definition of the weight concentration.

(7) For the Figure 4, please define the variable names in the two axes. I am not sure that “viscosity” and “temperature” will be better than “h” and “T”. As a rule, editors allow using such abbreviations. Let us wait the decision of technical editors.

(8) For the Figure 13, please define the variable names in the two axes. The same as before.

(9) For the Figure 15, please define the variable names in the two axes. The same as before.

(10) In line 428, please define the variable name "ε/kT". Corrected.

(11) For the Figure 17, please define all the variable names and "gear symbol" used in the Figure 17. Also add the axes names in the two axes. This is not “gear”, but “asterisk” that means the initial state of the dope. Concerning the axes names the same as before.

(12) For the Figure 18, please add the axes names in the two axes. The same as before.

(13) For the Figure 20, please add the axes names in the two axes. Also please define all the variable names and "STAGE 1", "STAGE 2" used in the Figure 20, to connective with "at the first stage" in line 522, "In the second stage" in line 523. Corrected and the same as before.

(14) In the Figure 2, please add the stage names for the three stages of schematic representations. There are two stages only: destroying intermolecular H-bonds and intramolecular H-bonds.

(15) In the Figure 1, please define the name "n". This figure is replaced.